# Prediction of the Near-Infrared Absorption Spectrum of Single-Walled Carbon Nanotubes Using a Bayesian Regularized Back Propagation Neural Network Model

**Takao Onishi** [1,*]**, Yuji Matsukawa** [1]**, Yuto Yamazaki** [1] **and Daisuke Miyashiro** [1,2,*]

[1] Department of Physics, Tokyo University of Science, 1-3 Kagurazaka, Shinjuku-ku, Tokyo 161-8601, Japan; 2214101@alumni.tus.ac.jp (Y.M.); scarlet0devil@gmail.com (Y.Y.)

[2] ESTECH Corp., 2-7-31 Fukuura, Kanazawa-ku, Yokohama 236-0004, Japan

\* Correspondence: T.Ohnishi80462@outlook.jp (T.O.); A29919@rs.tus.ac.jp (D.M.)

**Abstract:** DNA-wrapped single-walled carbon nanotubes (DNA-SWCNTs) in stable dispersion are expected to be used as biosensors in the future, because they have the property of absorption of light in the near infrared (NIR) region, which is safe for the human body. However, this practical application requires the understanding of the DNA-SWCNTs' detailed response characteristics. The purpose of this study is to predict, in detail, the response characteristics of the absorption spectra that result when the antioxidant catechin is added to oxidized DNA-SWCNTs, from a small amount of experimental data. Therefore, in the present study, we predicted the characteristics of the absorption spectra of DNA-SWCNTs using the Bayesian regularization backpropagation neural network (BRBPNN) model. The BRBPNN model was trained with the catechin concentration and initial absorption peaks as inputs and the absorption spectra after catechin addition as outputs. The accuracy of the predicted absorption peaks and wavelengths after the addition of catechin, as predicted by the BRBPNN model, was within 1% of the error of the experimental data. By inputting the catechin concentrations under hundreds of conditions into this BRBPNN model, we were able to obtain detailed prediction curves for the absorption peaks. This method has the potential to help to reduce the experimental costs and improve the efficiency of investigating the properties of high-cost materials such as SWCNTs.

**Keywords:** Bayesian regularization backpropagation neural network; near-infrared absorption; single-walled carbon nanotubes

## 1. Introduction

Carbon nanotubes (CNTs), which have excellent mechanical properties, are well-known for their use as composites for mechanical materials and as reinforcing materials for bodies such as aircraft and automobiles, as well as concrete [1]. Single-walled carbon nanotubes (SWCNTs) with a diameter of several nanometers, which are made by winding a graphene sheet in a cylindrical shape, are expected to be used as the next-generation of electronic materials, because they show the characteristics of conductors and semiconductors by a winding method called chirality. Generally, SWCNTs' powder contains different types of chirality during the manufacturing process. SWCNTs have the unique property of absorbing and emitting light in the near infrared (NIR) region, the exact wavelength that is emitted in which differs depending on the chirality. In particular, SWCNTs, which have absorption characteristics in the NIR region that is safe for the human body, are expected to be used for drug delivery and biosensing [2–4]. Focusing on the bio-application of SWCNTs, it is necessary that they show stable dispersibility in water for a long time. Zheng et al. [5] and Nakajima et al. [6] established a technology that can disperse SWCNTs in water more stably than conventional surfactants by wrapping the SWCNTs with DNA. Umemura's group reported that adding catechin antioxidants to oxidized DNA-SWCNTs

could result in a return to the NIR absorption spectrum that was present pre-oxidation and they also found that there was the possibility of using DNA-SWCNTs as biosensors to discriminate various molecules [7]. The optical spectral characteristics of SWCNTs depending on the electronic state of the SWCNTs' surfaces has been experimentally and theoretically investigated [8]. Although the theoretical mechanism is still unclear, it is thought that one of the explanations for the change in spectral characteristics when catechin is added to DNA-SWCNTs is that the spectral characteristics change due to the change in the electronic state of SWCNT's surface, due to redox [9]. In this way, although research is underway to investigate the optical properties by adding various molecules to SWCNTs [10], there are innumerable combinations of organic molecules that are optimal for the purpose, so many experiments are required. On the other hand, many studies using large-scale models, such as SWCNT and DNA-SWCNT composites, have been reported using the simulation approach [11–13]. However, it is difficult to calculate the various experimental conditions involved, such as catechin concentration and pH. Therefore, in the future, it is expected that a method will be devised that can accurately predict the results of different experimental conditions from a small amount of experimental data. Machine learning may be considered as one of the methods that may realize this possibility.

Machine learning uses various algorithms such as regression, classification, and deep learning. Deep learning (also known as machine learning), as represented by neural networks (NN), has been one of the most researched technologies since the middle of the 20th century and the rapid development of computational power [14]. Since the introduction of big data technologies [15], NN technology has significantly widened its fields of application in the last decade. NN technology has now become a mainstream methodology for image recognition [16,17], speech recognition [18], reinforcement learning [19], energy engineering [20], weather forecasting [21], disaster prediction [22], modelling environmental problems [23], and medical applications [24–26]. In recent years, the amount of research focused on applying machine learning to experiments on CNTs, a field which is on the frontiers of material science and engineering, has been increasing. For instance, Le has predicted the tensile strength of CNT composites using the Gaussian process regression model [27]. T. Hajilounezhad et al. were able to generate SWCNTs' forest morphology images using machine learning and their characterized forest synthesis-structure [28]. Yang et al. succeeded in predicting the recognition sequence of DNA from machine learning using the absorption spectra of DNA-SWCNTs that were stably dispersed in water using DNA [29]. These results are effective and innovative in comprehensively and quantitatively understanding the data, but they require a large amount of experimental data. In order to apply machine learning widely to the study of CNT materials, it is necessary to first consider case studies that can be applied even with a small amount of data.

In this study, we considered the development of a deep learning model that can accurately predict experimental data from a small amount of training data. The data targeted by the NN model in the present study was from an experiment on the NIR absorption spectrum change of a DNA-SWCNT dispersion, reported by Matsukawa et al. [30]. It is known that the absorption peaks of chirality (8,4)/(9,4) SWCNTs decreases when $H_2O_2$, which is an oxidizing agent, is added to the DNA-SWCNTs' dispersion [31]. (8,4)/(9,4) SWCNTs indicate that the absorption wavelengths for chirality (8,4) and (9,4) are almost the same. Matsukawa et al. have suggested the possibility of a biosensor that detects antioxidant activity because the peak amplitude is restored before oxidation when catechin, which has antioxidant properties, is added to oxidized DNA-SWCNTs, as shown in Figure 1. In order to put DNA-SWCNTs to practical use in biosensor applications, it is necessary to clarify the absorption spectra responses to detailed catechin concentration changes through more experiments. However, preparing many of these experiments is time consuming, costly and impractical. Thus, predictions with high accuracy, using machine learning from a small amount of experimental data, will contribute to efficient experiment planning in the future.

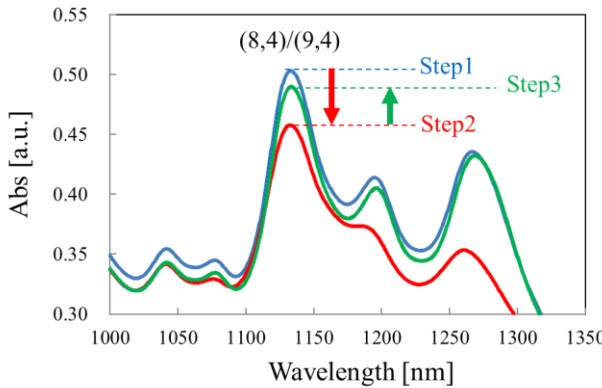

**Figure 1.** Typical reaction example of experimental data targeted by the BRBPNN model. Blue line of step 1 is NIR spectrum in the initial state of DNA-SWCNTs. Red line is NIR spectrum with $H_2O_2$ added to the initial state. Green line is NIR spectrum with catechin added in step 2.

## 2. Materials and Methods

### 2.1. Sample Data

This section shows the sample of experimental data that was used for training and verification by the NN model. The SWCNTs were purchased from Raymor Industries Inc. (Boisbriand, QC, Canada). The double strand DNA (deoxyribonucleic acid sodium salt from salmon testes, D1626) was obtained from Sigma-Aldrich Co. LLC (St. Louis, MO, USA). Hydrogen peroxide (abt. 30%, 084-07441) and catechin (553-74471) were purchased from Wako Pure Chemical Industries, Ltd. (Osaka city, Osaka, Japan). A 1 mg/mL DNA solution was prepared with 10 mM tris (hydroxymethyl) aminomethane-HCl (Tris-HCL) buffer (pH 7.9). To untangle the DNA molecules, the solution was sonicated in a bath-type ultrasonicator (80 W) for 90 min on ice. Finally, the dsDNA solutions were gently shaken for 3 h. For preparation of the DNA-SWCNTs, 0.5 mg of SWCNT powder and 1 mL of the DNA stock solution were mixed and sonicated for 2 h using a probe-type sonicator (3 W, VCX-130, Sonic & Materials, Inc., Newtown, CT, USA) on ice. The supernatant of the prepared DNA-SWCNT dispersion was stored, after centrifugation at 17,360 g for 3 h at 8 °C. Then, 0.5 mg of SWCNTs were suspended in 1 mL of the DNA solution. The samples were sonicated using the probe-type sonicator (3 W) for 2 h, followed by centrifugation at 17,360 g for 3 h at 8 °C. The supernatants were collected as a DNA–SWNT suspension.

The absorption spectrum measurement of the DNA-SWCNTs was carried out using the same procedure as detailed in the previous studies [30]. The absorption measurement (700–1350 nm) was taken using a NIR spectrometer (SolidSpec-3700DUV, Shimadzu Corporation, Kyoto City, Kyoto, Japan). For NIR measurements of DNA-SWCNTs, 100 μL of the DNA-SWCNT suspension and 880 μL of the Tris-HCL buffer solution were mixed in a cuvette and the initial spectra were recorded. In the second step, 10 μL of $H_2O_2$ that was diluted with sterilized water (with a final concentration of 0.03%) was added to the samples, followed by incubation for 30 min at 21 °C. The spectra of the samples were measured. Finally, 10 μL of catechin solution (with final concentrations at 15, 1.5, 0.15, 0.075, and 0.03 μg/mL) was added to the samples and the spectra were measured after 10 min

incubation at 21 °C. Triplicate NIR measurements for each experiment were recorded in order to verify the reproducibility.

## 2.2. Bayesian Regularized Backpropagation Neural Network

Machine learning was performed using the MATLAB Neural Network toolbox. The neural network model that was used for function approximation was a two-layer feedforward network that uses a sigmoid transfer function for the hidden layer and a linear transfer function for the output layer. In addition, the number of neurons in the hidden layer was set to 10 layers. The training function used a Bayesian regularized backpropagation neural network (BRBPNN) model. A typical neural network is trained for the purpose of minimizing the squared error, $E_D$, between the correct data and the predicted output. The Levenberg–Marquardt method is widely applied as a method for solving the minimum value of a nonlinear equation [32]. The BRBPNN model follows the optimization of the Levenberg–Marquardt method to generate a network that minimizes the combined function, $F$, of the squared error, $E_D$, and weight, $E_w$, that are shown in the following equation in order to generate a neural network that achieves proper fitting.

$$F = \beta E_D + \alpha E_w \tag{1}$$

where $E_w$ is the sum of squares of the neural network weights and $\beta$ and $\alpha$ are objective function parameters.

A technique for applying Bayesian regularization to neural network optimization was proposed by David MacKay [33] and applied to practical problems by Foresee and Hagan et al. [34]. The Bayesian framework considers network weighting factors to be random variables. After the data is retrieved, the weight density function is expressed as follows, according to Bayes' theorem.

$$P(w|D, \alpha, \beta, M) = \frac{P(D|w, \beta, M)P(w|\alpha, M)}{P(D|\alpha, \beta, M)} \tag{2}$$

where $D$ represents the dataset, $M$ is the specific neural network model that was used, and $w$ is the vector of network weights. $P(w|\alpha, M)$ is the pre-density and represents knowledge about the weights before the data was collected. $P(D|w, \beta, M)$ is a likelihood function, which is the probability that data will occur given the weight, $w$. $P(D|\alpha, \beta, M)$ is a normalizing coefficient with a total probability of 1. If we assume that the noise in the training set data is Gaussian, the probability densities can be written as follows.

$$P(w|\alpha, M) = \frac{1}{Z_w(\alpha)} exp(-\alpha E_w) \tag{3}$$

$$P(D|w, \beta, M) = \frac{1}{Z_D(\beta)} exp(-\beta E_D) \tag{4}$$

$$Z_w(\alpha) = \left(\frac{\pi}{\alpha}\right)^{\frac{N}{2}} \tag{5}$$

$$Z_D(\beta) = \left(\frac{\pi}{\beta}\right)^{\frac{n}{2}} \tag{6}$$

By substituting Equations (3) and (4) into Equations (2), the following relationship can be obtained:

$$P(w|D, \alpha, \beta, M) = \frac{\frac{1}{Z_w(\alpha)Z_D(\beta)} exp(-(\beta E_D + \alpha E_w))}{P(D|\alpha, \beta, M)} \tag{7}$$

When Equation (7) is expressed by Equation (1) of the objective function, it is expressed by the following equation:

$$P(w|D, \alpha, \beta, M) = \frac{1}{Z_F(\alpha, \beta)} exp(-F(w)) \tag{8}$$

In this Bayesian framework, optimal weights should maximize posterior probabilities; $P(w|D, \alpha, \beta, M)$. Maximizing posterior probabilities is equivalent to minimizing regularized objective functions.

### 2.3. Input/Output Data and the Verification Method

The BRBPNN model learns the relationships between input and output data so that you can identify them, as shown in Figure 2. The input data is the absorption of the (8,4)/(9,4) peaks in the initial state spectra of DNA-SWCNTs and various catechin concentration (15, 1.5, 0.15, 0.075 or 0.030 μg/mL). The output data was the NIR spectra after catechin addition, as shown in Figure 1. From the three experimental data captured for each catechin concentration, two data were used to train the model and the remaining one datum was used to validate the training model. The learning accuracy of the model was evaluated by comparing the prediction results that were obtained from the model with the rest of the validation data.

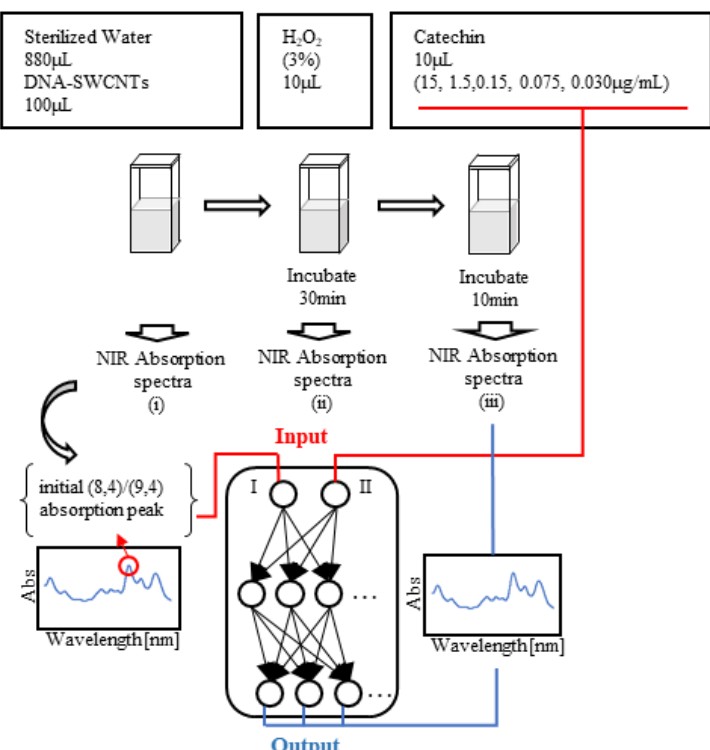

**Figure 2.** Overview of the learned BRBPNN model. Initial (8,4)/(9,4) peak absorption and catechin concentration are used as input data I, II, and NIR spectra after catechin addition in (iii) state as output data.

## 3. Results and Discussion

Figure 3 shows the results of comparing the predicted data and the verification data of the spectra after the catechin addition when the absorption peaks of (8,4)/(9,4) DNA-SWCNTs were in the initial state and the catechin concentration was input to the learned BRBPNN model. It can be confirmed that the spectral waveform accurately corresponds to a wide range of 700 to 1350 nm. The prediction results from the learning model represent the changes to the spectrum in response to the changes in the catechin concentration over a

wide range of 500-fold from 0.03 to 15 μg/mL. As is clear from Figure 3, there is almost no change in the absorption spectrum, even if the catechin concentration is diluted 5 times from 0.15 to 0.03 μg/mL. However, in order to understand the overall characteristics of DNA-SWCNTs as biosensors, it is necessary to predict not only the region that responds to changes in catechin concentration, but also the region that the reaction saturates. The significance of predicting the changes of absorption spectra with catechin concentration is that it leads to the possibility of estimating the amount of catechin molecules that affect the electronic state of the surface of DNA-SWCNTs. Hence, this may be useful for forming a hypothesis about the mechanism. Therefore, to validate our proposed approach, we constructed a BRBPNN model trained with DNA-SWCNTs' absorption spectral data for a wide range of catechin concentrations. The Levenberg–Marquardt method is often used as a learning algorithm that is faster than the BRBPNN model that was used in this study, but our experimental data could not predict a smooth spectral waveform due to noise in the waveform. Thus, the BRBPNN model, which is based on the Bayesian framework, is considered to be suitable for accurately predicting complex spectral waveforms such as those targeted in this study. In general, the computational speed and optimized accuracy of the neural network model are highly dependent on the hidden layer. In our research, as a result of creating a training model with 1 to 14 hidden layers and comparing them, we found that there was no significant improvement in the calculation speed and accuracy with 10 layers or more, so we decided to use 10 layers.

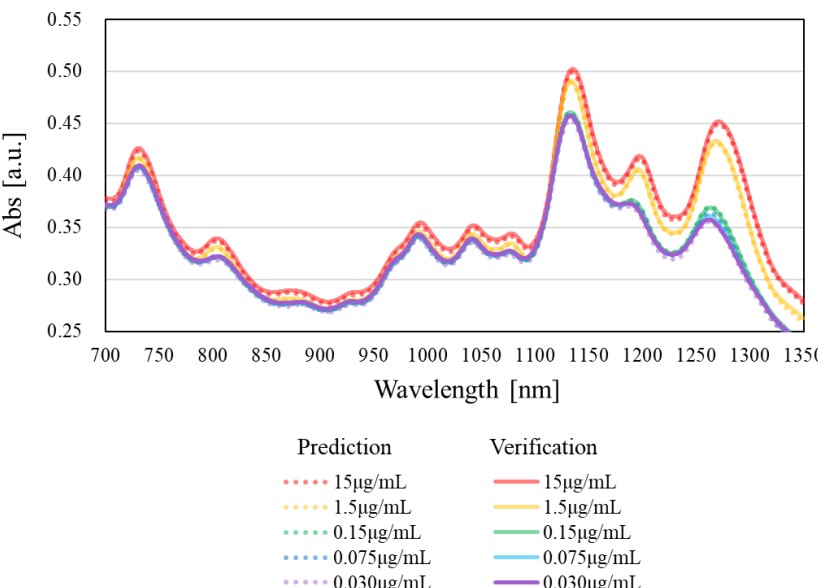

**Figure 3.** Comparison of the results of prediction data and verification data in the spectrum waveform after catechin addition for each catechin concentration.

The data used for training the BRBPNN contains errors because it is experimental data. Therefore, in order to investigate the effect of the data errors used for training, we performed cross-validation in which the training data and the verification data were exchanged. The spectral waveform after catechin addition is predicted from the three BRBPNN outputs created for each catechin concentration and the absorption and wavelengths of the (8,4)/(9,4) peaks are shown in Figure 4a,b. It was confirmed that the error of the prediction result is about the same as that of the verification data with which it was compared. This result is considered to indicate that, even if the experimental data that trains the neural network has some variation, the prediction accuracy of the BRBPNN model is not significantly affected and a highly robust result can be obtained. The average ± standard deviation of the absorption and wavelength at each catechin concentration, as shown in Figure 4, is also shown in Table 1.

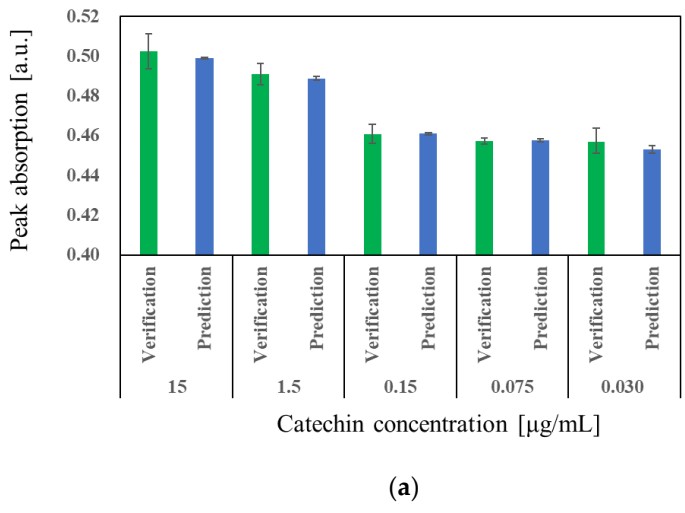

(**a**)

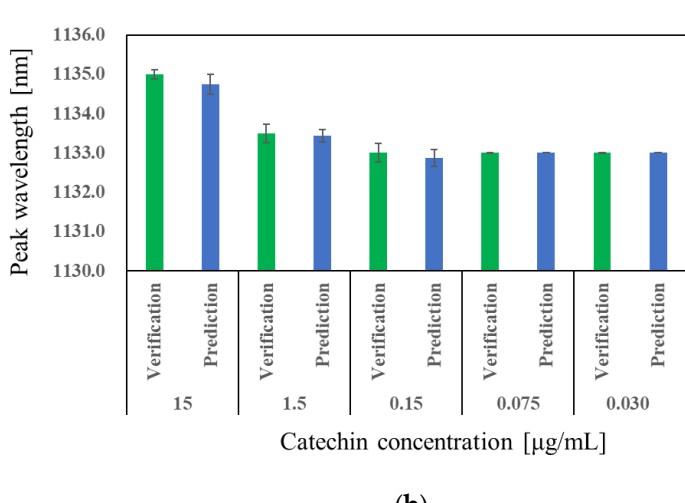

(**b**)

**Figure 4.** (**a**) Comparison of prediction data and verification data of absorption peak of (8,4)/(9,4) DNA-SWCNTs. (**b**) Comparison of prediction data and verification data of peak wavelength of (8,4)/(9,4) DNA-SWCNTs. Each value is expressed as mean ± standard deviation.

**Table 1.** Absorption and wavelength of (8,4)/(9,4) peak in prediction data and verification data.

| Input Data | | | Output Data | |
|---|---|---|---|---|
| Catechin Concentration [μg/mL] | (8,4)/(9,4) Peak | | (8,4)/(9,4) Peak | |
| | Absorption [a.u.] | Data Type | Absorption [a.u.] | Wavelength [nm] |
| 15 | 0.513 | Verification | 0.502 ± 0.009 | 1135.0 ± 0.1 |
| | | Prediction | 0.499 ± 0.000 | 1134.8 ± 0.3 |
| 1.5 | 0.499 | Verification | 0.491 ± 0.005 | 1133.5 ± 0.2 |
| | | Prediction | 0.489 ± 0.001 | 1133.4 ± 0.2 |
| 0.15 | 0.500 | Verification | 0.461 ± 0.005 | 1133.0 ± 0.2 |
| | | Prediction | 0.461 ± 0.001 | 1132.9 ± 0.2 |
| 0.075 | 0.498 | Verification | 0.457 ± 0.001 | 1133.0 ± 0.0 |
| | | Prediction | 0.458 ± 0.001 | 1133.0 ± 0.0 |
| 0.030 | 0.500 | Verification | 0.457 ± 0.006 | 1133.0 ± 0.0 |
| | | Prediction | 0.453 ± 0.002 | 1133.0 ± 0.0 |

It was confirmed that the BRBPNN model learned, using the data of the five conditions of catechin concentration, to accurately predict the NIR spectrum after catechin addition, as

shown in Table 1. We also investigated the prediction curves for the (8,4)/(9,4) absorption peaks for various catechin concentrations using the BRBPNN model. In order to draw this prediction curve, the catechin concentration input to the BRBPNN model was set to 370 patterns in the range of 0.005 to 15 μg/mL. In addition, the initial (8,4)/(9,4) absorption peaks were fixed at 0.5 because the fluctuation was small, as shown in Table 1. Figure 5 shows the results of extracting the (8,4)/(9,4) absorption peaks from the NIR spectrum output by inputting various catechin concentrations into the BRBPNN model. The blue circles in Figure 5 mark the results predicted by the BRBPNN model that trained with the experimental data at the total catechin concentration. This curve was reasonable in that it satisfied the error range of the verification data, not only for the catechin concentrations between 0.15 and 1.5 μg/mL, which has a large change in absorption peak, but also for the catechin concentrations between 0.03 and 0.15 μg/mL, which hardly change. It is clear that the black dotted line (approximated to the power of the verification data) and the gray dotted line (approximated to the polynomial) are not sufficiently fitted when compared to the prediction results of the BRBPNN model. These results suggest the possibility of predicting the NIR absorption spectra under hundreds of conditions from a small number of experimental data. Furthermore, we examined similar predictions using a model that was trained by omitting the data of specific concentration conditions from the training data of the five catechin concentration conditions and then investigated the sensitivity of the trained data to the prediction results. The light blue circles mark the prediction results from the BRBPNN model that learned from a dataset that omitted the catechin concentration of 0.075 μg/mL. Since the change in this result is small compared to the results before and after omitting it, it was confirmed that the prediction accuracy is almost the same even if there are four catechin concentration data to be learned from. On the other hand, the orange circles mark the results that were predicted using the BRBPNN model when it learned from a dataset that omitted the catechin concentration of 1.5 μg/mL from all the data. As a result, although the verification data of 0.03 μg/mL and 0.15 μg/mL are accurately expressed, the tendency of the experiment could not be grasped as a whole. This result shows that it is important to determine the conditions for training in order to predict the desired characteristics.

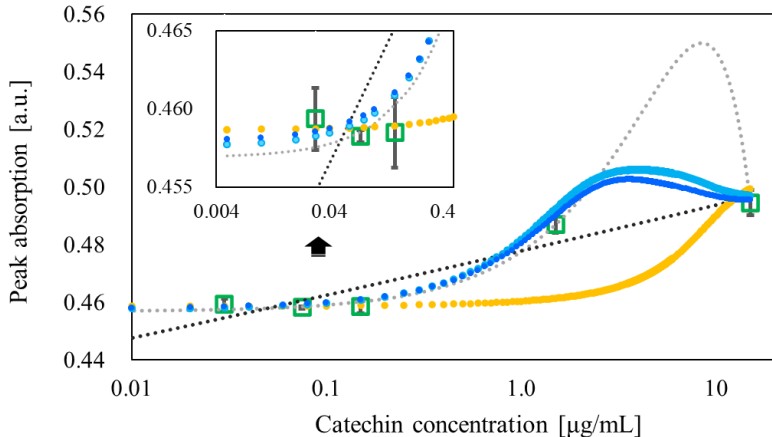

**Figure 5.** Prediction results of (8,4)/(9/4) absorption peaks for catechin concentration input using BRBPNN: The blue circles represent the BRBPNN model's output from learned data. The orange circles represent the BRBPNN model's outputs when the learned excluding the catechin 1.5 μg/mL. The light blue circles represent the BRBPNN model's output when learned excluded the catechin 0.075 μg/mL. The green squares, black lines and gray dotted lines represent the validation data, the power approximation lines and the polynomial approximation lines (quadratic) for them, respectively.

This study found that the BRBPNN model, based on the Bayes framework, can accurately predict complex waveform data, such as NIR absorption spectra. We also suggest that the input data that is given during training is very important when expressing the phe-

nomenon to be predicted by deep learning. In this paper, we constructed a deep learning model from experimental data in order to predict the desired spectra. Since the deep learning model is determined by the relationship between input and output, it is thought that a similar prediction model can be created for tannic acid and caffeine [9] as they are both a molecule with antioxidant activity that is separate from catechin, which was the focus of this study. We have already reported experimental studies showing similar absorption spectral changes with tannic acid as a molecule that exhibits antioxidant activity against DNA-SWCNTs [35]. In addition, this method can be applied not only to the absorption spectrum, but also to the waveform data of other experiments. For example, many studies have been reported in which the force curve was measured by directly contacting the mechanical properties of the surfaces of biomolecules under various physiological conditions, using a cantilever of an atomic force microscope [36,37]. It is known that it is difficult to discriminate data under different conditions because measurement of biomolecules in water is unstable. The deep learning model presented in this study may also help to identify those data easily. Utilizing deep learning models for understanding and improving the efficiency of experiments is considered to be an important initiative for promoting research on carbon materials in the future.

## 4. Conclusions

We have developed a BRBPNN model based on the Bayesian framework that can accurately predict changes in the NIR absorption spectrum when catechins with antioxidant activity are added to a DNA-SWCNT dispersion. The BRBPNN model was trained with the catechin concentrations and initial (8,4)/(9,4) absorption peaks as inputs and the absorption spectra after catechin addition as outputs. The predicted values of the absorption peaks and wavelengths in (8,4)/(9,4) chirality after the addition of catechin by the model were within 1% of the error compared to the experimental verification data. By inputting the catechin concentration of 300 conditions or more into the BRBPNN model developed from a small set of five conditions, we were able to obtain a detailed prediction curve for the (8,4)/(9,4) absorption peaks. These prediction curves were reasonable curves that satisfied the error range of the verification data, regardless of the magnitude of the absorption peak change, with respect to the change in catechin concentration. These results could not be represented by typical approximation curves. Neural network models such as the BRBPNN model may be useful for planning efficient experiments by making overall predictions from a small amount of data in advance when clarifying the physical characteristics of high-cost materials such as SWCNTs. In addition, this method can accurately predict the relationship between input and output data, thus it can contribute to the prediction of various experimental data, not limited to absorption spectra.

**Author Contributions:** Conceptualization, T.O. and D.M.; methodology, T.O., Y.M. and Y.Y.; software, T.O. and Y.Y.; validation, T.O., Y.Y. and D.M.; formal analysis, T.O.; investigation, T.O. and D.M.; resources, Y.M.; writing—original draft preparation, T.O. and D.M.; writing—review and editing, D.M.; visualization, T.O. and D.M.; supervision, D.M. All authors have read and agreed to the published version of the manuscript.

**Funding:** This research received no external funding.

**Data Availability Statement:** The description of the training function used in this study is provided by The MathWorks, Inc. as follows: https://www.mathworks.com/help/deeplearning/ref/trainbr.html?searchHighlight=trainbr&s_tid=srchtitle_trainbr_1 (accessed on 20 November 2021).

**Acknowledgments:** The authors thank the permission to use the MATLAB license provided by Tokyo University of Science.

**Conflicts of Interest:** The authors declare no conflict of interest.

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
