# Peer review of "Prediction of the Near-Infrared Absorption Spectrum of Single-Walled Carbon Nanotubes Using a Bayesian Regularized Back Propagation Neural Network Model"

_carbon_

Round 1
Reviewer 1 Report
The main purpose of the paper is to present results of using a machine learning (ML) based model for prediction of CNT absorption at a wavelength of 1.3 microns. More specifically the relation of the absorption and the chemical modification of the CNT surface with catechin is discussed.
The topic by itself is important since getting idea on the chemical composition via simple optical experiment is a promising possibility.
Still the paper does not prove that the model or whatever approach it describes is of any practical use. I leave aside the details of the algorithms used for machine learning implementation. Based on the description of the results of the modeling and its experimental verification it remains unclear
- What CNTs where used in the experiment. What is meant by ”chirality (8,4) / (9,4) SWCNTs”? A mixture of two chiralities? Then what was the ratio between the two? What was the reason for choosing these two specific chiralities combined with the wavelength of 1.3 microns?
- As the catechin concentration is changed by three orders of magnitude the absorption changes by about 10% which is compared to the experimental error. What is the original motivation for using this technique for characterizing the catechin concentration?
- Will the change in the wavelength range allow for better efficiency?
The second point is of biggest importance since it questions the very point of carrying out the entire study. Before it is explained properly by the authors the paper cannot be of interest to the broad readership of the journal.
Author Response
Response to reviewer #1:
We are grateful for your helpful comments and useful suggestions that have helped us to improve our paper. According to your comments, we modified the manuscript as follows:
- First, the reason for choosing the absorption bands of (8,4)/ (9,4) chirality as the target of prediction is that the absorption peak is the most prominent and it is easy to handle as a sensor. Then, commonly manufactured SWCNTs contain various types of chirality, and it is often the case that one absorption wavelength contains multiple chirality [2]. In addition, the chirality ratio cannot be accurately determined, but it does not seem to affect the prediction of antioxidant activity in this study. Thus, we added it below in the revised manuscript.
>>see from p.1 l.35 to p.1 l.38.
>>see from p.2 l.80 to p.2 l.81.
- The motivation for this study is to predict in detail the response characteristics of the absorption spectrum when antioxidant catechins are added to oxidized DNA-SWCNTs from a small amount of experimental data, which will be useful for the development of practical biosensors. In this study, we proposed a deep learning model that can predict a wide range of catechin concentrations from a small amount of data. Thus, we added it below in the revised manuscript.
>> see from p.9 l.304 to p.9 l.306.
- If the change in the peak such as 1270 nm is clear, it is thought that the same prediction accuracy as the BRBPNN model developed in this study can be shown. This is because the deep learning model is determined by the relationship between data input and output. We added it below in the revised manuscript.
>>see from p.8 l.261 to p.8 l.264.
Reviewer 1:
The main purpose of the paper is to present results of using a machine learning (ML) based model for prediction of CNT absorption at a wavelength of 1.3 microns. More specifically the relation of the absorption and the chemical modification of the CNT surface with catechin is discussed.
The topic by itself is important since getting idea on the chemical composition via simple optical experiment is a promising possibility.
Still the paper does not prove that the model or whatever approach it describes is of any practical use. I leave aside the details of the algorithms used for machine learning implementation. Based on the description of the results of the modeling and its experimental verification it remains unclear
- What CNTs where used in the experiment. What is meant by ”chirality (8,4) / (9,4) SWCNTs”? A mixture of two chiralities? Then what was the ratio between the two? What was the reason for choosing these two specific chiralities combined with the wavelength of 1.3 microns?
- As the catechin concentration is changed by three orders of magnitude the absorption changes by about 10% which is compared to the experimental error. What is the original motivation for using this technique for characterizing the catechin concentration?
- Will the change in the wavelength range allow for better efficiency?
The second point is of biggest importance since it questions the very point of carrying out the entire study. Before it is explained properly by the authors the paper cannot be of interest to the broad readership of the journal.

Reviewer 2 Report
The study on prediction of near-infrared absorption spectrum of single-walled carbon nanotubes using the BRBPNN model is interesting and worth pursuing. The authors developed a BRBPNN model based on the Bayesian framework which can accurately predict changes in the NIR absorption spectrum when catechins with antioxidant activity are added to the DNA-SWCNTs dispersion. The calculation data were compared with the experimental data, showing that the developed model fits the experimental data well. The manuscript is well written, and the results are well presented and discussed.
Author Response
Response to reviewer #2:
We greatly appreciate your comment and evaluation. We brushed up the manuscript and added citations to clarify the novelty of our work without changing the essential conclusions as shown below.
- Based on the comments of other reviewers, we added it below in the revised manuscript.
>>see from p.1 l.35 to p.1 l.38.
>>see from p.2 l.80 to p.2 l.81.
>>see from p.8 l.262 to p.8 l.265.
>>see p.3 l.107
>>see p.6 l.185
>>see p.8 l.231
>>see from p.8 l.239 to p.8 l.240.
>>see from p.8 l.261 to p.8 l.264.
>> see from p.9 l.304 to p.9 l.306.
Reviewer 2:
The study on prediction of near-infrared absorption spectrum of single-walled carbon nanotubes using the BRBPNN model is interesting and worth pursuing. The authors developed a BRBPNN model based on the Bayesian framework which can accurately predict changes in the NIR absorption spectrum when catechins with antioxidant activity are added to the DNA-SWCNTs dispersion. The calculation data were compared with the experimental data, showing that the developed model fits the experimental data well. The manuscript is well written, and the results are well presented and discussed.

Reviewer 3 Report
The article presents the predictions regarding the characteristics of the absorption spectrum of DNA-SWCNTs, obtained when using the Bayesian regularization backpropagation neural network (BRBPNN).
Results obtained indicate the accuracy of the main absorption peak and wavelength after the addition of catechin predicted by the BRBPNN in comparison to the experimental data. The authors used the neuronal network predictions by inputting different catechin concentrations and as result, they obtained detailed prediction curves for the main absorption peak. The proposed method has the potential to help reduce experimental costs and improve efficiency when investigating the properties of high-cost materials such as SWCNTs. The presentation is clear, the language is adequate and the results are presented in a concise form. For this reason, I recommend publishing this article.
Author Response
Response to reviewer #3:
We are grateful for your helpful comments and useful suggestions that have helped us to improve our paper. According to other reviewer’s comments, we modified the manuscript as follows:
>>see from p.1 l.35 to p.1 l.38.
>>see from p.2 l.80 to p.2 l.81.
>>see from p.8 l.262 to p.8 l.265.
>>see p.3 l.107
>>see p.6 l.185
>>see p.8 l.231
>>see from p.8 l.239 to p.8 l.240.
>>see from p.8 l.261 to p.8 l.264.
>> see from p.9 l.304 to p.9 l.306.
Reviewer 3:
The article presents the predictions regarding the characteristics of the absorption spectrum of DNA-SWCNTs, obtained when using the Bayesian regularization backpropagation neural network (BRBPNN).
Results obtained indicate the accuracy of the main absorption peak and wavelength after the addition of catechin predicted by the BRBPNN in comparison to the experimental data. The authors used the neuronal network predictions by inputting different catechin concentrations and as result, they obtained detailed prediction curves for the main absorption peak. The proposed method has the potential to help reduce experimental costs and improve efficiency when investigating the properties of high-cost materials such as SWCNTs. The presentation is clear, the language is adequate and the results are presented in a concise form. For this reason, I recommend publishing this article.

Reviewer 4 Report
In this work, authors report on the use of Bayesian regularized back propagation neural network model for the prediction of N-IR absorption spectrum of SWCNTs. The purpose of this study is to predict in detail the response characteristics of the absorption spectrum when antioxidant catechin is added to oxidized DNA-SWCNT from a small amount of experimental data. This is a well written, interesting and to the point paper and all conclusions drawn are supported by the data used and the model developed. I only have a few minor comments regarding this manuscript.
- A few typos need to be corrected, line 104, leave space between “ultrasonicator” and (80 W) and between parenthesis and “for”. Line 182, represent not represents, lines 260-264, this is a very large sentence and due to grammar mistakes it is difficult to understand what the authors claim here, lines 236 to 237, the sentence needs to be rewritten, it is confusing like that. Line 228, leave space between table and 1.
- Lines 102-103 please explain why you used this buffer Tris-HCl pH 7.9 in order to make the DNA solution. The use of PBS or HEPES would make a difference?
- Did the authors use for comparison purposes another molecule with antioxidant activity to test their model?
Author Response
Response to reviewer #4:
We greatly appreciate your comment and evaluation. We brushed up the manuscript and added citations to clarify the novelty of our work without changing the essential conclusions as shown below.
- We have corrected all the typos pointed out by the reviewers and rewrote the lines shown below.
>>see p.3 l.107
>>see p.6 l.185
>>see p.8 l.231
>>see from p.8 l.239 to p.8 l.240.
>>see from p.8 l.268 to p.8 l.271.
- The reason for using buffer Tris-HCl pH 7.9 was used for comparison with previous studies. In addition, we can't answer because we haven't experimented with PBS and HEPES.
- There are experimental studies using tannic acid as another antioxidant. Although a BRBPNN model has not been created for these experimental data, the effect of this tannic acid on the absorption spectrum of SWCNTs is similar to that of catechin, thus, it can be expected that the same results as in this study will be obtained. we added it below in the revised manuscript.
>>see from p.8 l.261 to p.8 l.266.
Reviewer 4:
In this work, authors report on the use of Bayesian regularized back propagation neural network model for the prediction of N-IR absorption spectrum of SWCNTs. The purpose of this study is to predict in detail the response characteristics of the absorption spectrum when antioxidant catechin is added to oxidized DNA-SWCNT from a small amount of experimental data. This is a well written, interesting and to the point paper and all conclusions drawn are supported by the data used and the model developed. I only have a few minor comments regarding this manuscript.
- A few typos need to be corrected, line 104, leave space between “ultrasonicator” and (80 W) and between parenthesis and “for”. Line 182, represent not represents, lines 260-264, this is a very large sentence and due to grammar mistakes it is difficult to understand what the authors claim here, lines 236 to 237, the sentence needs to be rewritten, it is confusing like that. Line 228, leave space between table and 1.
- Lines 102-103 please explain why you used this buffer Tris-HCl pH 7.9 in order to make the DNA solution. The use of PBS or HEPES would make a difference?
- Did the authors use for comparison purposes another molecule with antioxidant activity to test their model?

Round 2
Reviewer 1 Report
The authors did not respond properly to my criticism, see details in the attached file

Author Response
Response to reviewer #1:
Thank you for the additional explanation. According to your comments, we modified the manuscript as follows:
Does your claim mean that the absorption peak change for catechin concentrations from 0.003 to 0.15 is so small that it does not act as a sensor?
The theme of this manuscript is to show how to accurately predict experimental data with a small amount of data. Therefore, as a biosensor, we aim to express not only the region where the absorption peak change with respect to the catechin concentration is large, but also the region where the absorption peak change is saturated with the learning model.
>>see from p.6 l.185 to p.6 l.193.
>>see from p.8 l.259 to p.8 l.262.
>>see from p.10 l.324 to p.10 l.325.
>>see from p.10 l.329 to p.10 l.331.
Although we received a comment that Given lack of novelty in terms of algorithms or approaches to ML, there is no research report that applied BRBPNN to the absorption spectrum of DNA-SWCNT and proved a model with high prediction accuracy with a small amount of data.
In addition, in order to show novelty, we also state in the manuscript that the traditional algorithm, Levenberg-Marquardt method, cannot accurately predict the absorption spectrum.
>>see from p.6 l.193 to p.6 l.196.
Reviewer 1:
My original comment 2.:
As the catechin concentration is changed by three orders of magnitude while the absorption changes by about 10% which is compared to the experimental error. What is the original motivation for using this technique for characterizing the catechin concentration? REPSONSE: The motivation for this study is to predict in detail the response characteristics of the absorption spectrum when antioxidant catechins are added to oxidized DNA-SWCNTs from a small amount of experimental data, which will be useful for the development of practical biosensors. In this study, we proposed a deep learning model that can predict a wide range of catechin concentrations from a small amount of data. Thus, we added it below in the revised manuscript.
>> see from p.9 l.304 to p.9 l.306.
It will also contribute 304 to the development of biosensor applications that can detect antioxidant effects such as 305 catechins.
Here the authors do not seem to understand my point at all. I try to explain it once again. When you want to know the quantity A (i.e. concentration of antibodies or some chemical in the sample) but cannot measure it directly you may find another quantity B (say, resistance or transmission at a certain wavelength), that is easier to measure and is know to DEPEND on the quantity A. In this case you make a SENSOR: a device in which the quantity A can be found by measuring quantity B. Before you use the sensor, you need to calibrate it somehow. Another, more advanced, option is when instead of measuring SINGLE quantity B you measure a set of values {Bi}, i.e., absorptions at different wavelengths. In this case ML can be used a lieu the calibration. In BOTH cases it is crucial that B or {Bi} DO depend on A In the case considered in the paper the change in quantity A (catechin concentration) by THREE ORDERS of magnitude results in FEW PERCENT change in the peak absorption. For anyone understanding real applications and sensing it means that B does NOT depend on A. That means that the entire work described in the paper although mathematically correct has no practical use. Given lack of novelty in terms of algorithms or approaches to ML, I don’t see why the paper should ever be published.
